# Current Evidence of the Role of the Myokine Irisin in Cancer

**DOI:** 10.3390/cancers13112628

**Published:** 2021-05-27

**Authors:** Evangelia Tsiani, Nicole Tsakiridis, Rozalia Kouvelioti, Alina Jaglanian, Panagiota Klentrou

**Affiliations:** 1Department of Health Sciences, Brock University, St. Catharines, ON L2S 3A1, Canada; nt16rw@brocku.ca (N.T.); rkouvelioti@brocku.ca (R.K.); nb_ajaglanian@brocku.ca (A.J.); 2Centre for Bone and Muscle Health, Brock University, St. Catharines, ON L2S 3A1, Canada; nklentrou@brocku.ca; 3Department of Kinesiology, Brock University, St. Catharines, ON L2S 3A1, Canada

**Keywords:** exercise, contraction, muscle, health benefits, myokines, irisin, cancer, in vitro, in vivo

## Abstract

**Simple Summary:**

Regular exercise/physical activity is beneficial for the health of an individual and lowers the risk of getting different diseases, including cancer. How exactly exercise results in these health benefits is not known. Recent studies suggest that the molecule irisin released by muscles into the blood stream after exercise may be responsible for these effects. This review summarizes all the available in vitro/cell culture, animal and human studies that have investigated the relationship between cancer and irisin with the aim to shed light and understand the possible role of irisin in cancer. The majority of the in vitro studies indicate anticancer properties of irisin, but more animal and human studies are required to better understand the exact role of irisin in cancer.

**Abstract:**

Cancer is a disease associated with extreme human suffering, a huge economic cost to health systems, and is the second leading cause of death worldwide. Regular physical activity is associated with many health benefits, including reduced cancer risk. In the past two decades, exercising/contracting skeletal muscles have been found to secrete a wide range of biologically active proteins, named myokines. Myokines are delivered, via the circulation, to different cells/tissues, bind to their specific receptors and initiate signaling cascades mediating the health benefits of exercise. The present review summarizes the existing evidence of the role of the myokine irisin in cancer. In vitro studies have shown that the treatment of various cancer cells with irisin resulted in the inhibition of cell proliferation, survival, migration/ invasion and induced apoptosis by affecting key proliferative and antiapoptotic signaling pathways. However, the effects of irisin in humans remains unclear. Although the majority of the existing studies have found reduced serum irisin levels in cancer patients, a few studies have shown the opposite. Similarly, the majority of studies have found increased levels of irisin in cancer tissues, with a few studies showing the opposite trend. Clearly, further investigations are required to determine the exact role of irisin in cancer.

## 1. Introduction

Cancer is the leading cause of death worldwide, with an estimated 10 million deaths in 2020 [1]. This upward trend in global cases and mortality is influenced by potentially modifiable behaviors that are linked to cancer, such as smoking, poor diet, alcohol consumption and lack of exercise [2]. Cancer cells display enhanced growth rates and a resistance to programmed cell death termed apoptosis. In healthy cells, homeostasis and the maintenance of normal tissue architecture is tightly controlled by cellular signaling cascades. In cancer cells, mutations in key signaling molecules lead to homeostatic dysregulation [3,4,5,6,7,8,9,10]. Growth factor-initiated signaling cascades are activated when growth factors bind to their receptors. These cascades are dysregulated in cancer cells, leading to enhanced proliferation and inhibition of apoptosis [5,6,7,8,9,10,11]. Enhanced growth factor signaling may be due to increased growth factor production, receptor mutations or mutations in downstream key signaling molecules. 

The phosphatidylinositol 3-kinase (PI3K)/protein kinase B (Akt) pathway is heavily involved in cellular growth and homeostasis. This pathway is often mutated in many types of cancers and is implicated in deregulated cellular proliferation and apoptosis. Mutated/activated Akt leads to tumorigenesis, as well as resistance to chemotherapy [12,13,14,15] and radiotherapy [16,17]. 

The mammalian target of rapamycin (mTOR) and the ribosomal S6 kinase (p70 S6K) are activated downstream of Akt, leading to increased protein synthesis and proliferation [5,7,18,19,20,21]. Activated Akt also leads to enhanced survival and the inhibition of apoptosis [5]. Another signaling molecule that is frequently mutated/activated in cancer is Ras [10]. Activated Ras leads to the activation of mitogen-activated protein kinases (MAPK), resulting in tumorigenesis and chemo and radiation resistance [8]. 

Initiation, promotion and progression are the three stages involved in the development of cancer [22]. Normal, healthy cells acquire a series of hallmark capabilities, such as sustaining proliferation, evading growth suppressors and resisting cell death, which enable them to become malignant [3,4]. 

Regular physical activity is associated with many health benefits and a reduced risk of many cancers, including lung, prostate and breast cancer [23,24,25,26,27,28,29,30,31,32,33]. Women who exercise regularly were found to have a significantly decreased cancer risk [26,30,31,34]. In addition, frequent exercise was found to increase the survival of breast cancer patients [35]. Similarly, regular physical activity was shown to reduce the risk of lung adenocarcinomas in men and women [23,24,29]. Despite this epidemiological evidence, the exact mechanisms involved in the anticancer effects of regular physical activity/exercise are not known.

Recent evidence indicates that the exercising muscle releases proteins called myokines into the bloodstream, allowing them to be delivered to different tissues in the body and exert the beneficial effects of exercise (Figure 1). Prominent myokines produced by skeletal muscle are interleukins-6 and -15 (IL-6 and IL-15), oncostatin, myostatin, brain-derived neurotrophic factor (BDNF) and irisin [36,37,38].

The present review summarizes the available in vitro and in vivo studies focusing on irisin and cancer. The key words: irisin, cancer, in vitro, in vivo, exercise and anticancer were searched using the PubMed database. The studies are presented chronologically, and summary data tables are included. It should be noted that a number of related past reviews exist [39,40,41,42,43,44,45,46,47,48]; however, the present review provides updated information and adds to the existing literature. 

## 2. Irisin

### 2.1. Irisin Structure and Synthesis

Irisin, a recently discovered myokine, was named after the ancient Greek messenger goddess Iris, whose role was to deliver messages to the gods living on Mount Olympus. The protein released by the exercising muscle was thought to play a similar role in delivering messages to tissues in the body and bringing them the beneficial effects of exercise and therefore justifying the naming. Irisin is part of the fibronectin type III domain containing 5 (FNDC5) protein. The human FNDC5 gene has a different start codon (ATA) compared to other species such as the mouse or rat (ATG) [49], and this ATA start codon is associated with a low expression efficiency [50]. The reader is recommended to see the recent reviews and studies by Maak et al. [48] and Albrecht et al. [51,52] for more information and the controversies surrounding irisin expression and detection. In a recent study, Albrecht et al. found that, from the three annotated human FNDC5 transcripts (T1, T2, T3), only one was expressed in human skeletal muscle [52]. Two additional transcripts (T4,T5) were found, indicating FNDC5 transcript diversity in human tissues.

Muscle cells produce fibronectin type III domain containing 5 (FNDC5), a protein whose destination is to be localized to the plasma membrane. The original FNDC5 protein contains an N-terminal signal sequence (Figure 2), which targets it to the plasma membrane, and is subsequently cleaved, as is true for all signal peptide sequences of plasma membrane proteins. The N-terminal signal sequence is followed by a fibronectin type III domain (FNIII), a transmembrane domain and a c-terminal tail corresponding to the cytosolic region of the protein (Figure 2). Irisin is produced following the proteolytic cleavage of the mature FNDC5 protein (Figure 2). The resulting N-terminal fragment, corresponding to the extracellular FNDC5 ectodomain, is known as irisin and is released into the circulation [53,54]. Irisin is a 112-amino acid peptide with a molecular weight of 12 kDa [53].** The amino acid sequence of irisin is highly conserved in all sequenced mammalian species, and mouse and human irisin are 100% identical.

The increased expression of FNDC5 and irisin in response to muscle contractions is thought to be mediated by the transcriptional coactivator peroxisome proliferator-activated receptor-γ coactivator 1α (PGC1α) [53,55]. However, evidence has shown that PGC1α is a transcriptional coactivator and does not bind to DNA itself but, instead, interacts with transcription factors to effect the gene expression [56]. The orphan nuclear receptor estrogen-related receptor alpha (ERRα, also known as NR3B1) is a central metabolic regulator [57,58] and was found to interact with PGC1α to regulate the FNDC5 gene expression [59,60,61]. The interaction between ERRα and PGC1α has been studied in skeletal muscle and suggests that PGC1α alone is not sufficient to regulate the FNDC5 gene expression. Muscle contractions lead to an increase in intracellular calcium and activation of the calcium-dependent calmodulin kinase (CaMK), as well as the protein phosphatase calcineurin. Different transcription factors, including the cAMP response element-binding protein (CREB), nuclear factor of activated T cells (NFAT), myocyte enhancer factor 2 C (MEF2C) and -D (MEF2D), are activated in response to increased calcium signaling leading to the increased expression of PGC1α (Figure 2B). In addition, muscle contractions reduce the cellular ATP levels, resulting in an increased AMP/ATP ratio and activation of the metabolic energy sensor AMP-dependent kinase (AMPK). AMPK directly phosphorylates PGC1α at threonine-177 and serine-538 [62]. PGC1α phosphorylation leads to the increased expression of FNDC5 and irisin (Figure 2B) [53,63,64,65]. It has also been reported that AMPK activation by other stimuli such as icariin increased the expression of FNDC5 /irisin in C2C12 murine skeletal cells in vitro [66]. Post-translational modification includes the N-glycosylation of FNDC5 and irisin [67,68,69]. Two N-glycosylation sites located at the Asn-7 and Asn-52 residues of irisin have been identified and early evidence indicates an important role of glycosylation on the biological activity and function [67].

### 2.2. Irisin Blood Levels, Clearance and Tissue Distribution

Following the discovery of irisin by Spiegelman’s group [53], several subsequent studies have examined the levels of irisin in humans. The measurements indicated that, in a sedentary individual, the irisin levels are at approximately 3.6 ng/mL, while the levels found in individuals after exercise are much higher at ~4.3 ng/mL [68]. Similarly, Moreno et al. [70] found higher circulating irisin levels in physically active individuals compared to sedentary subjects. Different studies have reported different levels of irisin in human blood, ranging from ng/mL [68] to μg/mL [70]. The reason for these discrepancies may be due to the utilization of different detection assays and clearly indicates the requirement of more research. Most of the studies measuring irisin levels in human serum use enzyme-linked immunoassay (ELISA), and the validity of these assays has been questioned, as there is the possibility of nonspecific binding of the irisin antibodies to other plasma/serum proteins [51,52]. Mass spectrometry is considered the gold standard for the detection and quantification of irisin, and unfortunately, only a few studies have utilized it to measure irisin in human serum.

Overall, the existing data provided evidence that irisin is released in response to exercise. The half-life of irisin was examined in vivo by Kim et al. in mice injected with recombinant irisin for 6 days and was found to be less than an hour [71]. If these data extend to humans, and the half-life of endogenously produced irisin in humans is also less than an hour, they could contribute to the variability seen in the blood/circulating irisin levels. Apart from exercise, cold exposure and various drugs such as statins, metformin and phytochemicals/polyphenols such as resveratrol and citrus flavonoids, as well as diet, could all influence the blood irisin levels [47].

The tissue distribution and clearance of irisin in vivo has not been extensively studied. Lv et al. administered radioactively labelled irisin by injection in mice and utilized single-photon emission computerized tomography (SPECT)/CT imagining to examine irisin distribution in different tissues. The highest level of irisin was found in the gallbladder, followed by the liver and kidneys [72]. It was also found that the clearance of irisin is through the hepato-biliary and the renal system [72].

Aydin et al. using immunohistochemistry (IHC), found irisin in the skeletal muscles, peripheral nerves, ovaries/testes, pancreas, liver, spleen and stomach [73]. Additional studies have shown expression in adipose, heart, kidney, lung, prostate, intestine and thymus tissue [47]. As mentioned above, the specificity of the existing irisin antibodies is questionable [51,52], it is possible that the antibodies used in IHC or Western blotting bind to other proteins, and these controversies require more research to be resolved.

### 2.3. Irisin receptor and Mechanism of Action in Target Tissues

As mentioned above, irisin is a type I transmembrane messenger protein that is mainly secreted by the skeletal muscle and stimulates “browning” and thermogenesis in fat tissue [53] and induces changes in different tissues in response to exercise [39,44]. Currently, there is no irisin receptor identified. Recent studies have shown that, in osteocytes, irisin binds to the αV/β5 integrin receptor complex and activates the focal adhesion kinase (FAK), a well-established downstream signaling molecule of a αV/β5 integrin receptor, while the use of αV/β5 integrin receptor inhibitors blocked the irisin effects [71]. Similarly, using the immunofluorescence analysis, it was shown in vitro and in vivo that irisin binds to the integrin αVβ5 receptor on gut epithelial cells [46,74]. Wei et al. [75] found that the use of integrin (αV) inhibitors (RGD peptide, Echistain) blocked the irisin effects on hepatocytes (L02), proving further evidence of irisin signaling via integrin receptor binding. These recent studies provided the initial evidence of the binding of irisin to αVβ5 integrin complexes. However, it is possible that irisin does not bind solely on this complex and may bind to other members of integrins or other membrane receptors. Further studies are needed in the future for a better understanding of irisin receptors. 

## 3. Role of Irisin in Cancer

### 3.1. Role of Irisin in Cancer: In Vitro Evidence 

Limited studies (presented below) have examined the direct effects of irisin on cancer cells. The exposure of endometrial (KLE and RL95-2), colon (HT2 and, MCA38), thyroid (SW579 and BHP7) and esophageal (OE13 and OE33) cancer cells to physiological (5–10 nmol/L) and high physiological/pharmacological (50–100 nmol/L) concentrations of irisin did not affect cell adhesion and colony formation [76] (Table 1). These data indicated no changes in the proliferation and malignant potential of cancer cells within irisin treatment. 

In contrast to the above findings, the exposure of MDA-MB-231 breast cancer cells to irisin (0.625–20 nM for 24 h) resulted in a significant reduction in cell viability and migration without affecting the nonmalignant MCF-10a breast epithelial cells [77] (Table 1). In addition, the treatment with irisin increased caspase-3/7 cleavage, increased apoptosis and suppressed NF-κB activation in MDA-MB-231 breast cancer cells. Furthermore, irisin, in combination with doxorubicin, a chemotherapeutic drug widely used in the treatment of breast cancer, resulted in enhanced effects. These data indicate that irisin inhibits the proliferation and induces the apoptosis of breast cancer cells and has the potential to act as a chemosensitizer, enhancing the effectiveness of doxorubicin. Likewise, the viability of androgen receptor-positive (LNCaP) and androgen receptor-negative (DU-145 and PC3) human prostate cancer cells was significantly decreased by the treatment with irisin (10–100 nM) [78].

The treatment of A549 and NCI-H446 lung cancer cells with irisin resulted in a significant inhibition of cell proliferation, migration and invasion and inhibited the epithelial-to-mesenchymal transition (EMT) [79] (Table 1). Furthermore, the treatment with irisin inhibited the PI3K/AKT pathway, a key regulator of Snail, and reduced the expression of Snail [79]. These data indicated the strong anticancer effects of irisin in lung cancer cells [79]. Additionally, the real-time polymerase chain reaction (RT-PCR) analysis showed that NCI-1703 and NCI-H522 lung cancer cells had a higher expression of FNDC5 mRNA and higher levels of irisin protein, detected by Western blots and immunofluorescence, compared to normal lung fibroblasts [80]. However, a recent study by Fan et al. [81] found lower FNDC5 mRNA levels in A549, H1299, H358 and H1650 non-small cell lung cancer (NSCLC) cells compared to normal lung epithelial cells. The treatment of lung cancer cells with 20-nM irisin for 24–96 h suppressed their proliferation whereas silencing of FNDC5 enhanced their proliferation [81]. The increased expression of FNDC5 in lung cancer cells blocked nuclear factor-κB (NF-κB) activation and downregulated the multidrug resistance protein 1 (MDR1) levels [81]. Furthermore, it was found that NSCLC cells that were resistant to paclitaxel, a chemotherapeutic agent used in the treatment of lung cancer, had decreased FNDC5 levels, and irisin treatment was able to attenuate this chemoresistance [81].

The exposure of osteosarcoma cells to irisin resulted in significantly reduced proliferation, migration and invasion [82]. Furthermore, treatment with irisin abolished the IL-6-induced epithelial–mesenchymal transition (EMT) and the IL-6-stimulated phosphorylation of the signal transducer and activator of transcription 3 (STAT3) and downstream induction of Snail [82]. The use of a STAT3 inhibitor enhanced the effect of irisin on the EMT and Snail expression [82]. These data indicate that irisin prevents the IL-6-induced EMT of osteosarcoma cells via inhibition of the STAT3/Snail signaling pathway.

Cheng et al. investigated the effects of irisin on human osteosarcoma U2OS cells [83]. Irisin dose-dependently inhibited the viability, migration and invasion of osteosarcoma cells. MicroRNA miR-214-3p inhibited irisin/FNDC5 expression and promoted migration, epithelial/mesenchymal transition (EMT) and the invasion of U2OS cells [83] (Table 1). This study suggests a potential of targeting irisin and miR-214-3p towards the treatment of osteosarcoma.

The treatment of HepG2 and SMCC7721 hepatocellular carcinoma cells with irisin resulted in increased cell proliferation, migration and invasion [84], and these effects were inhibited by the PI3K inhibitor (LY294002). In addition, the cytotoxic effect of doxorubicin in HepG2 cells were decreased by irisin treatment. These data indicate that irisin via the activation of the PI3K/AKT pathway may increase liver cancer progression and decrease the sensitivity to chemotherapy [84].

Pancreatic cancer cells (MIA PaCa-2 and Panc03.27) had significantly reduced survival and growth and induced G1 cell cycle arrest in response to the treatment with recombinant irisin [85]. The migration of pancreatic cancer cells was significantly reduced with irisin treatment. EMT and vimentin expression were reduced, while E-cadherin expression was increased [85]. Irisin increased the phosphorylation/activation of the energy sensor AMPK and inhibited the activity of the mammalian target of rapamycin (mTOR), a key player in protein synthesis and cell proliferation. These data indicate the anticancer effects of irisin that are mediated by the AMPK- mTOR-signaling cascade [85]. The treatment of pancreatic cancer cells (MIA PaCa-2 and Panc03.27) with irisin in combination with the chemotherapy drugs doxorubicin or gemcitabine resulted in an enhanced response to chemotherapy [86]. Doxorubicin-induced apoptosis was increased with irisin, and this was associated with increased poly (ADP-ribose) polymerase (PARP) and caspase-3 cleavage, and the reduced expression of B-cell lymphoma/lewkmia-2 (BCL-2) and B-cell lymphoma-extra-large (BCL-xL) [86]. In addition, irisin reduced the phosphorylation/activation of Akt and reduced NF-κB. These data show the important anticancer effects of irisin in pancreatic cancer cells that are mediated by the inhibition of the PI3K/Akt/ NF-κB signaling cascade [86].

Similar results were obtained when the PANC-1 and BxPC-3 human pancreatic cancer cells were treated with irisin [87] (Table 1). A dose-dependent inhibition of cell proliferation and induction of apoptosis was seen with the irisin treatment [87]. The migration and invasion of these pancreatic cancer cells were significantly reduced with the irisin treatment. These effects were associated with the inhibition of Akt phosphorylation/activation [87].

In addition, Yang et al. examined the effects of irisin on erastin-induced ferroptosis in pancreatic cancer PANC-1 cells. Ferroptosis, is regulated cell death/apoptosis due to an iron-dependent increase in ROS cellular levels. Irisin was found to enhance the erastin-induced increase in free iron, total ROS levels and glutathione depletion [88]. PANC-1 cells express endogenous irisin and its removal utilizing the small interference RNA approach (siRNA targeting FNDC5) and abolished the erastin-induced effects [88]. The reduced NRF2 (nuclear reacting factor 2) and p62 levels were seen in a combined erastin–irisin treatment [88]. It should be noted that the effects of irisin alone on the iron, ROS, NRF2 levels and cell viability were not significant. Overall, this study indicates a potential of irisin to enhance erastin-induced ferroptosis in pancreatic cancer cells. 

Huang et al. exposed U-87 MG, T98G and LN-18 glioblastoma cells to 1-μM irisin and found a significant inhibition of cell proliferation that was associated with a G2/M cell cycle arrest and increased levels of p21 [89] (Table 1). The irisin treatment also inhibited cell invasion by upregulating the mRNA and protein expression of tissue factor pathway inhibitor-*2 (*TFPI-2). The coculture of glioma (U-87 MG) cells with adipocytes (3T3L1) induced an aggressive phenotype, as seen by their enhanced invasion ability. However, the coculture of glioma cells with irisin-treated adipocytes reduced the invasion, indicating a reversal of this aggressive phenotype [89]. An examination of the culture media revealed reduced levels of several invasion-related proteases in the media, from glioma cells cocultured with irisin-treated adipocytes compared to the media from glioma cells cocultured with adipocytes. These reduced levels of the invasion-related proteases, responsible for extracellular matrix (ECM) degradation, may explain the reduced invasive ability of glioma cells. These in vitro data raise the possibility that adipocytes surrounding tumors may influence cancer cell aggressiveness in vivo.

Overall, the evidence from the majority of the available in vitro studies indicates the inhibition of proliferation, survival, migration and invasion and the induction of apoptosis of cancer cells exposed to irisin (Table 1). These anticancer effects of irisin were associated with the inhibition of PI3K, Akt, mTOR, STAT3 and NF-κB and activation of AMPK (Figure 3). It is not known if the inhibition of PI3K, Akt, BCL-2, NF-kB, STAT3 and mTOR by irisin is direct or indirect. It is possible that irisin directly inhibits them or indirectly modulates the upstream regulators. For example, the inhibition of mTOR may be due to the activation of the cellular energy sensor AMPK, an upstream regulator (inhibitor) of mTOR. 

The levels of cleaved caspase 3, 7 and PARP, all markers of apoptosis, were enhanced with the irisin treatment (Figure 3). It is important to note that one study showed no effects of irisin on cancer cell proliferation [76], while another study utilizing hepatocellular carcinoma cells showed increased proliferation, migration and invasion with irisin treatment [84] that was associated with activation of the PI3K/Akt signaling pathway. The existence of these contradictory evidence point to the requirement of more research and in-depth investigation of the biological effects and the role of irisin in tissue homeostasis. It may also suggest that the effects of irisin are cell- and tissue-specific.

### 3.2. Role of Irisin in Cancer: In Vivo Evidence 

The role of irisin in cancer has also been examined in vivo (in animals and in humans) (Table 2, Table 3 and Table 4). 

#### 3.2.1. Animal Models of Cancer

Altay et al. examined the levels of irisin in control BALB/c mice and in mice with gastric tumors induced by the administration of N-nitroso-N-methylurea (MNU) [90] (Table 2). The serum levels of the markers of inflammation and cachectic factors were elevated in mice with induced cancer compared to the control animals. FNDC5 and the cachectic factor zinc-α-2 glycoprotein were not detected in gastric tissues in any animal group. However, the levels of FNDC5 mRNA were increased in white and brown adipose tissue in mice with experimentally induced gastric cancer (precancer group mice with glandular stomach/adenomatous hyperplasia, and cancer group mice with intramucosal and invasive cancer) [90]. Whether these elevated levels of FNDC5 gene expression in white and brown adipose tissue contribute to cachexia is not clear, and more studies are required.

The administration of irisin (20 μg/day for 2 weeks) by injection into animals xenografted with U-87 MG human glioma cells resulted in reduced tumor volumes [89] (Table 2). Additionally, the use of radiolabeled irisin and micro-PET/CT-fused imaging showed a significant accumulation in the tumors, indicating a potential of irisin to target cancer cells in vivo [89]. Furthermore, slices of tumors were stained for adiponectin (used as a marker for adipocyte staining), and the data showed that the regions with a strong radioactive irisin signal had positive adiponectin staining, indicating that irisin and adipocytes were colocalized [89]. Interestingly, only adipocytes in tumors and areas of invasion exhibited an enhanced irisin uptake [89]. These data are very interesting and suggest that irisin may target the tumors in vivo by influencing/modulating cancer-associated adipocytes. 

#### 3.2.2. Human Studies

The serum irisin levels were found significantly lower in female patients with invasive ductal breast cancer compared with the control healthy women [91] (Table 3). A significant independent association between the serum irisin levels and development of breast cancer was found in these female patients. There was an estimation that a 1-μg unit of increase in the irisin levels results in almost a 90% decrease in the risk of breast cancer development. A significant association was also found between the irisin levels and tumor stages (stages I and III) in breast cancer female patients, with marginal associations between the irisin levels and tumor sizes and lymph nodes [91]. After adjusting for the effects of age and body mass index (BMI), significantly lower serum irisin levels were also found significantly lower in breast cancer female patients with spinal metastasis compared to those without spinal metastasis, suggesting a protective role of irisin on spinal metastasis in breast cancer patients [92]. 

In a study by Gaggini et al. it was found that the plasma irisin levels, measured by enzyme-linked immunosorbent assay (ELISA), did not differ between hepatocellular carcinoma (HCC) patients and the control group [93]. This indicates that there was no correlation between the hepatic FNDC5/irisin levels and plasma irisin levels. This lack of correlation may be explained by the fact that circulating irisin levels are produced in several tissues or the lack of reliability of the polyclonal antibody used in the study [93]. Shi et al. also found that, in 20 patients with HCC serum, the irisin levels were similar to control healthy patients [84] (Table 3).

Zhang et al. measured the serum irisin levels in healthy subjects and in patients with HCC before and after hepatectomy. The serum irisin levels were significantly reduced in the cancer patients [94]. Low serum irisin levels before hepatectomy were significantly correlated with high postoperative complications (high comprehensive complication index (CCI) scores) [94]. The authors suggested a potential of irisin to be used as a serum biomarker in the diagnosis of HCC and a predictor of the complications after hepatectomy. A similar study published recently by Pazgan-Simon et al. found significantly lower irisin serum levels in patients with HCC compared to the controls. Patients with a more advanced disease grade had the lowest irisin levels. Interestingly, lower irisin levels were seen in patients with reduced liver function/advanced cirrhosis. Overall, the irisin serum levels were inversely related to the severity of liver disfunction [95].

Similar results were reported in patients with other types of cancer. Zhu et al. examined the serum irisin levels in patients with colorectal cancer (CRC) and found reduced levels compared to the control healthy individuals [96]. After adjustment for age, gender, BMI and other biochemical parameters, it was found that subjects with high irisin levels had a lower (by 78%) risk for CRC compared to the risk in the subjects with low irisin levels. Altay et al. found increased serum irisin and carcinoembryonic antigen (CEA) levels in patients with renal cell cancer compared to the healthy controls [97]. These data suggest that irisin may be used as a diagnostic biomarker for renal cancer. 

Esawy et al. measured the serum irisin levels in healthy subjects and newly diagnosed bladder cancer patients and found significantly lower levels of irisin in cancer patients [98]. The serum irisin levels were positively correlated with the BMI and negatively correlated with the serum cholesterol levels [98]. Cancer patients with low serum irisin levels had increased mortality rates compared to patients with higher serum irisin levels.

The serum irisin levels were also found in significantly lower levels in prostate cancer patients compared with healthy subjects [99]. Finally, measurements of the serum irisin levels in healthy and newly diagnosed gastric cancer (GC) patients found significantly higher levels in gastric cancer patients compared to healthy controls, suggesting a possible role of serum irisin as a biomarker of GC disease diagnoses [100].

Panagiotou et al. [101] very recently found higher serum irisin levels (using ELISA) in women with both benign and malignant breast cancer tumors compared to healthy controls. The irisin serum levels were comparable in patients with neoplasms. Furthermore, the serum irisin levels were associated with tumor histological characteristics, including the Elston-Ellis grading system, estrogen receptors status and Ki67 levels [101]. This evidence suggests that the serum irisin levels may be used as a predictor of breast malignancy.

Aydin et al. examined the levels of irisin in various cancer tissues by IHC utilizing an irisin antibody and compared them to the levels found in control healthy tissues [102] (Table 4). All the tissues were from patients who received no chemotherapy or radiotherapy before their operations. Irisin was found in most of the tissues examined (brain, esophageal, stomach, liver, colon and pancreas), with significantly increased levels seen in gastrointestinal cancer and grade II astrocytoma tissues compared to the control [102].

Similarly, Kuloglu et al. examined by IHC the expression of irisin in breast, ovarian, cervical and endometrial healthy and cancerous tissues obtained by surgery from patients that had not received any chemotherapy or radiation treatment and had a sedentary lifestyle. Increased levels of irisin were found in breast, ovarian, cervical and endometrial tumor tissues compared to control healthy tissues [103] (Table 4).

The FNDC5 mRNA levels were found higher (tenfold) in HCC tissues from patients undergoing liver transplantation in comparison to deceased liver donor controls [93]. In HCC patients, the hepatic FNDC5/Irisin mRNA levels were positively correlated with the inflammation markers tumor necrosis factor-α and interleukin-6 mRNA, as well as with the mRNA of stearoyl-CoA desaturase (SCD-1), the main enzymatic regulator of de novo lipogenesis [93]. Based on these data of the enhanced expression of irisin in HCC tissues, in association with the enhanced expression of the genes involved in lipogenesis and inflammation, it is suggested by the authors that irisin may have a protective role against liver damage. These findings are in agreement with Shi et al. where the measured levels of irisin mRNA in the liver tissue of the patients with HCC were upregulated compared to the control healthy patients (sevenfold) [84]. 

Zhang et al. found that the expression of FNDC5/irisin in the tumor tissue of HCC patients was reduced [94] compared to the tissue from control healthy subjects.

In a study by Zhu et al. it was found that, when comparing the FNDC5 mRNA levels in colorectal cancer patients with or without obesity, there was no significant difference between the groups [96].

Additionally, a significant decrease in the irisin levels measured using IHC was also seen in chromophobe renal cell carcinoma (RCC) compared to the control healthy renal tissue [104]. Irisin was not detected in clear cell and papillary RCC, while the levels in oncocytoma were similar to the control [104].

Along those lines, examination of the expression of irisin in human healthy and thyroid cancer tissues revealed increased levels of irisin in oncocytic papillary carcinoma and anaplastic carcinoma tissues [105]. Nowinska et al. [80] found increased FNDC5 mRNA levels (measured by RT-PCR) and increased irisin protein levels (measured by Western blot and IHC) in human lung cancer cell lines NCI-1703 squamous cell carcinoma and NCI-H522 adenocarcinoma compared to IMR-90 healthy lung fibroblast cells. In addition, the FNDC5 gene expression was found in human non-small cell lung cancer (NSCLC, adenocarcinoma AC and squamous cell carcinoma SCC) tissues and stromal fibroblasts [80]. Higher grades of malignancy, a larger tumor size and lymph node metastasis were associated with a lower irisin expression in NSCLC cells and higher expression in stromal cells [80]. These data suggest a potential of irisin to be used as a prognostic factor for survival in NSCLC, with a high irisin expression in stromal cells indicating an aggressive/proliferative cancer and reduced survival. 

Overall, the evidence from the in vivo studies does not provide a clear picture of the exact role of irisin in cancer. The majority of the studies showed lower serum irisin levels in breast [91,92], hepatocellular carcinoma [94,95], colorectal [96], bladder [98] and prostate [99] cancer patients compared to healthy humans. Three studies found higher serum irisin levels in hepatocellular carcinoma [93], renal [97] and bladder [100] cancer patients compared to healthy humans (Table 3).

Examination of the irisin levels in cancer tissues have found increased expression in gastrointestinal [102], breast, ovarian, cervical and endometrial [103], HCC [94] and thyroid [105] cancer tissues and reduced expression in hepatocellular carcinoma [94,95], renal cell carcinoma [104] and NSCLC [80] tissues (Table 4). Irisin has been shown to increase the expression of uncoupling proteins (UCP1 in adipocytes) and increase thermogenesis [106]. The increased FNDC5/irisin expression in cancer tissues might be a compensatory mechanism to kill cancer cells. The increased irisin levels will increase thermogenesis, and local hyperthermia will lead to disruption of the cell cycle and induction of apoptosis. 

## 4. Conclusions

The majority of the available in vitro studies indicate the inhibition of cancer cell proliferation, survival and migration with irisin treatment. As mentioned in Section 2.1, irisin is post-translationally modified by glycosylation, and early evidence indicated an important role of this glycosylation process in the biological activity of irisin. Unfortunately, the majority of the published in vitro studies utilized a recombinant non-glycosylated irisin, and this could influence the data. Another important issue is that the irisin concentrations used in in vitro studies are in the nM range and much higher than the concentrations found in vivo. Future studies should resolve the concentration issue and examine the biological effects and the action of glycosylated irisin in different cancer cells. The cell and tissue distributions of the irisin receptors and the exact signaling cascades activated by irisin must be further explored. It is possible that the action of irisin is cell- and tissue-specific. 

In addition, the effects of irisin in human and animal cells in culture may be different from the effects seen in vivo. The establishment of global FNDC5 knockout mice by different groups, discussed in the recent review by Maak et al. [48], indicates no major problems as viability, weight, growth and fertility, are normal and no abnormalities were observed under standard conditions. The utilization of both global and tissue specific FNDC5 knockout mice in the future, will hopefully reveal more information on the role of FNDC5/irisin in cancer.

Studies from breast [91,92], hepatocellular [94], colorectal [96], renal [97], prostate [99] and gastric [100] cancer patients suggest that the serum irisin levels may serve as a diagnostic marker. However, the reliability and validity of the commercially available ELISA assays [48,51,52], utilized in most of the studies to measure irisin in human serum, are questionable and should be resolved. Hopefully, more studies utilizing mass spectrometry, the gold standard, will be performed in the future to assess irisin in the serum of healthy individuals and cancer patients. In addition, more research is required to determine the serum irisin glycosylation status in cancer patients. It is possible that both the glycosylation levels and total serum irisin levels may depend on what tissue/organ is cancerous. More studies are required to examine the changes in circulating irisin with the development of cancer before irisin can be implemented as a diagnostic biomarker.

Some in vivo studies have found increased irisin levels in various cancerous tissues, while others have shown the opposite. Furthermore, it is not clear whether the altered expression of irisin seen in tumor tissue is the cause of tumorigenesis or a compensatory mechanism to counteract tumorigenesis. 

Although the FNDC5/irisin expression was detected in various tissues, including healthy and neoplastic ones, controversies exist [48,51,52], and future studies should resolve them and provide a clearer picture of irisin expression. Many studies have utilized IHC to detect the irisin protein levels in healthy and tumor tissues. IHC involves the use of antibodies to detect a protein of interest (in this case, irisin) in samples/tissues that have been fixed. Other studies have utilized Western blotting to examine the irisin levels in different tissues. The irisin levels in different tissues, detected by IHC or Western blotting, could reflect endogenously/tissue expressed irisin or irisin that arrived at that specific tissue through the circulation. The studies that describe the detection of irisin in cancer tissues were performed utilizing tumor tissue removed from the patients by surgery. The handling of the tumor tissue, time to fixation and the specificity of the different antibodies used could affect the data. All the current controversies surrounding the detection of the irisin gene and protein expression in tissues should be resolved. Irisin expression in different cancer tissues should be studied extensively, and its role in tumorigenesis should be elucidated before irisin can be used for cancer diagnosis, prognosis and/or treatment. Clearly, more in vivo animal and human studies are required.

## Figures and Tables

**Figure 1 cancers-13-02628-f001:**
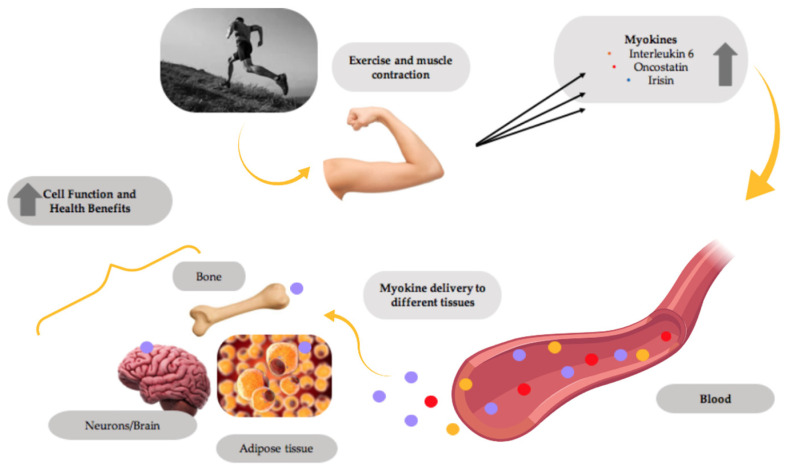
Myokines mediate the health benefits of exercise. Muscle contractions induce the production and release of myokines by muscle, which are delivered through the blood circulation to different tissues in the body, improving the overall tissue function and providing health benefits. This figure was created with BioRender.com.

**Figure 2 cancers-13-02628-f002:**
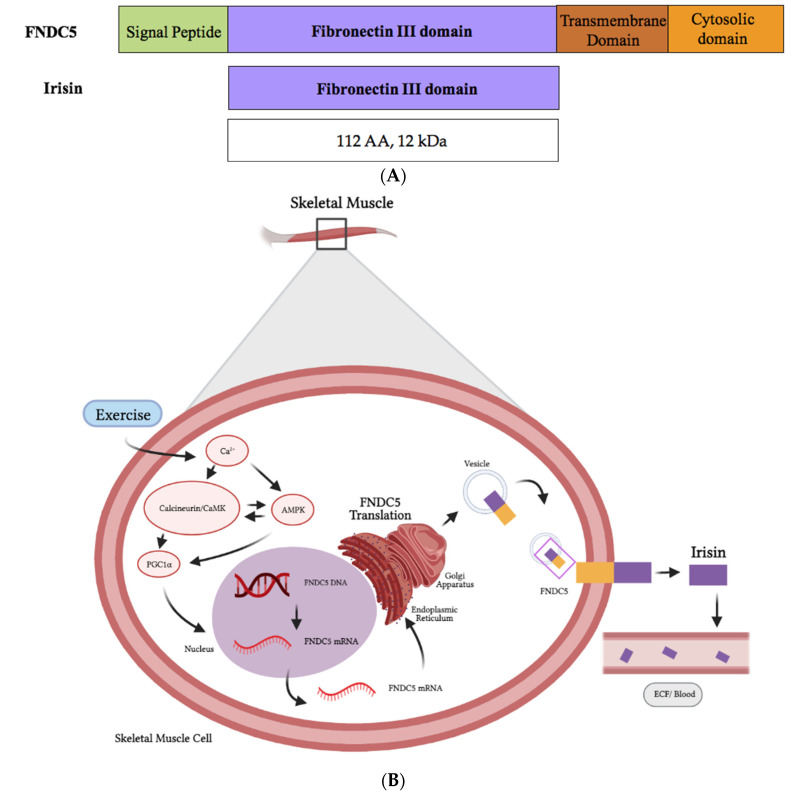
Schematic representation of FNDC5 and the irisin protein domains (**A**). The synthesis and release of irisin by muscle cells is induced by exercise (**B**). Figure 2B was created with BioRender.com.

**Figure 3 cancers-13-02628-f003:**
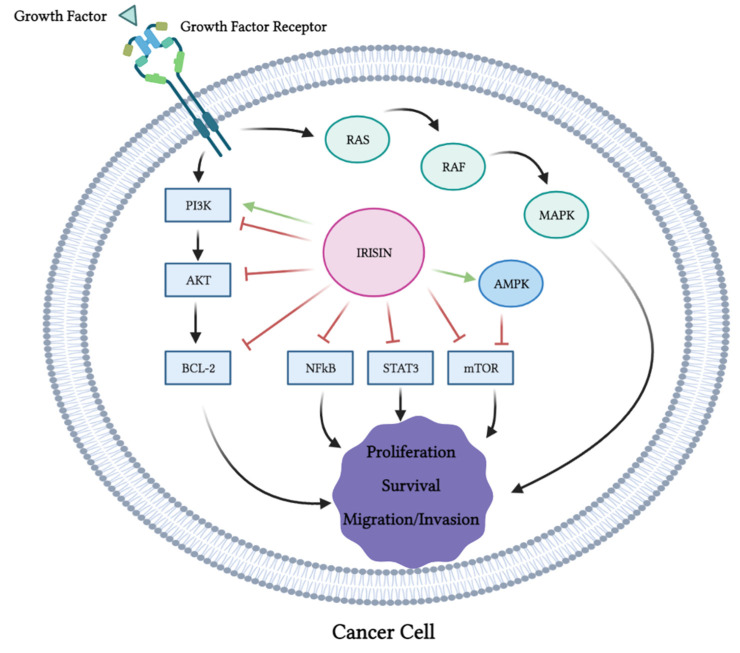
Effects of irisin on cancer cell signaling molecules. PI3K, Akt, Bcl-2, NF-κB, STAT3 and mTOR were all inhibited by irisin. Irisin activated AMPK and MAPK. This figure is based on the data of the studies mentioned in the in vitro section and Table 1. The figure was created with BioRender.com.

**Table 1 cancers-13-02628-t001:** Role of irisin in cancer: in vitro evidence.

Cancer Cell	Irisin Concentration/Duration	Findings	Reference
KLE, RL95-2 endometrial cancer HT29, MCA38 colon cancer SW579, BHP7 thyroid cancer OE13, OE33 esophageal cancer	5, 10, 50, 100 nM/36 h	no effect on cell adhesion no effect on colony formation	[76]
MDA-MB-231 breast cancer	0.625–20 nM/24 h	↓ cell viability ↓ cell migration ↓ NF-κB ↑ caspase 3/7cleavage ↑ apoptosis	[77]
LNCaP (androgen receptor positive) prostate cancer DU-145, PC3 (androgen receptor negative) prostate cancer	0.1, 1, 10 and 100 nM/24 h	↓ proliferation (cell viability)	[78]
A549, NCI-H446 lung cancer	20 nM/24 h	↓ proliferation ↓ migration ↓ invasion ↓ EMT ↓ PI3K/AKT ↓ Snail	[79]
A549, H1299, H358, H1650 lung cancer	20 nM/24–96 h	↓ proliferation ↓ MDR1 ↓ NF-κB	[81]
U2OS, MG-63 osteosarcoma	100 ng/mL/24 h	↓ proliferation ↓ migration ↓ invasion ↓ EMT ↓ STAT3 ↓ Snail	[82]
U2OS osteosarcoma	25, 50, 100, 200 ng/mL	↓ proliferation ↓ migration ↓ EMT ↓ invasion	[83]
HepG2, SMCC7721 hepatocellular carcinoma	2.5 nM/24 h	↑ proliferation ↑ migration ↑ invasion ↑ PI3K	[84]
MIA PaCa-2, Panc03.27 pancreatic cancer	10 and 100 nM/24 h	↓ proliferation ↓ colony formation ↓ migration ↑ cell cycle arrest (G1) ↓ EMT ↓ vimentin ↑ E-cadherin ↑ AMPK activation ↓ mTOR activation	[85]
MIA PaCa-2, BxPC-3, Panc03.27 pancreatic cancer	5, 10, 50, 100 nM/24 h	↑ apoptosis ↑ PARP cleavage ↑ caspase-3 cleavage ↓ BCL-2 ↓ BCL-xL ↓ Akt ↓ NF-κB	[86]
PANC-1,BxPC-3 pancreatic cancer	0, 10, 20 and 50 nM/24 h	↓ proliferation ↑ apoptosis ↓ migration ↓ invasion ↓ Akt	[87]
PANC-1 pancreatic cancer	100 nM/12 h	↑ erastin-induced apoptosis ↑ ROS levels ↓ GSH levels ↓ NRF2 ↓ P62	[88]
U-87 MG, T98G, LN-18 glioblastoma	1 μM/72 h	↓ proliferation ↑ cell cycle arrest ↑ p21 mRNA, protein ↓ invasion ↑ TFPI-2 mRNA, protein	[89]

**Table 2 cancers-13-02628-t002:** Role of Irisin in cancer: in vivo evidence from animal studies.

Animal Model	Intervention (Treatment)	Findings	Reference
BALB/c mice	N-nitroso-N-methylurea (MNU) to induce gastric cancer	↑ FNDC5 mRNA levels in white and brown adipose tissue of mice in pre-cancer and cancer groups ↑ Irisin levels in mice of pre-cancer and cancer groups	[90]
Athymic male nude mice injected with human glioblastoma cells (U-87 MG)	irisin 20 μg/day, 14 days	↓ reduced tumor volume	[89]

**Table 3 cancers-13-02628-t003:** Role of irisin in cancer: in vivo evidence from human studies measuring the serum irisin levels.

Participants	Measurements	Findings	Reference
Healthy humans, breast cancer patients (101 invasive ductal) *N* = 152	serum irisin levels (ELISA)	↓ serum irisin levels in breast cancer patients	[91]
Breast cancer patients (+ spinal metastasis) *N* = 148	serum irisin levels (ELISA)	↓ serum irisin levels in breast cancer patients with spinal metastasis	[92]
Hepatocellular carcinoma patients *N* = 36	serum irisin levels (ELISA)	no significant difference in serum irisin levels in HCC patients vs. donors	[93]
Patients with hepatocellular carcinoma *N* = 20	serum irisin levels (ELISA)	no significant difference in serum irisin levels	[84]
Healthy humans, hepatocellular carcinoma patients *N* = 219	serum irisin levels (ELISA)	↓ serum irisin levels in HCC patients ↓ FNDC5/irisin levels in HCC tissues	[94]
Healthy humans, hepatocellular carcinoma patients *N* = 43	serum irisin levels (ELISA)	↓ serum irisin levels in HCC patients ↓ FNDC5/irisin levels in HCC tissues	[95]
Colorectal cancer patients Obese and non-obese *N* = 116	serum irisin levels (ELISA)	↓ serum irisin levels in CRC patients	[96]
Renal cancer patients *N* = 48	serum irisin levels (ELISA)	↑ serum irisin levels in renal cancer patients	[97]
Healthy humans, newly diagnosed bladder cancer patients *N* = 150	serum irisin levels (ELISA)	↓ serum irisin levels in bladder cancer patients	[98]
Healthy humans, prostate cancer patients *N* = 80	serum irisin levels (ELISA)	↓ serum irisin levels in prostate cancer patients	[99]
Healthy humans, gastric cancer patients *N* = 51	serum irisin levels (ELISA)	↑ serum irisin levels in gastric cancer patients	[100]
Healthy humans, breast cancer patients *N* = 213	serum irisin levels (ELISA)	↑ serum irisin levels in both benign and malignant breast tumor cases compared to control	[101]

**Table 4 cancers-13-02628-t004:** Role of irisin in cancer: in vivo evidence from human studies measuring irisin in cancer tissues.

Participants	Measurements	Findings	Reference
Healthy humans, tumor tissues (brain, esophagus, stomach liver, pancreas) *N* = N/A	irisin expression in healthy and cancer tissues (IHC)	↑ irisin levels in gastrointestinal cancer, grade II astrocytoma	[102]
Healthy humans, tumor tissues (breast, cervix, ovaries, endometrium) *N* = N/A	irisin expression in healthy and cancer tissues (IHC)	↑ irisin levels in breast, ovarian, cervical and endometrial tumor tissues	[103]
Hepatocellular carcinoma patients *N* = 36	FNDC5 mRNA levels measured in liver tissues of HCC patients and controls (RT-PCR)	↑ FNDC5/irisin hepatic mRNA levels	[93]
Healthy humans, patients with hepatocellular carcinoma *N* = 20	FNDC5 mRNA levels measured in liver tissues of HCC patients and controls (RT-PCR)	↑ FNDC5 mRNA levels in HCC patients compared to controls	[84]
Healthy humans, hepatocellular carcinoma patients *N* = 219	FNDC5/irisin expression in HCC tissues	↓ FNDC5/irisin levels in HCC tissues	[94]
Healthy humans, hepatocellular carcinoma patients *N* = 43	FNDC5/irisin expression in HCC tissues	↓ FNDC5/irisin levels in HCC tissues	[95]
Colorectal cancer patients obese and non-obese *N* = 116	FNDC5/irisin levels in subcutaneous and visceral white adipose tissues (RT-PCR)	no difference between FNDC5 levels in subcutaneous and visceral white adipose tissues.	[96]
Renal cancer patients *N* = 110	irisin expression in healthy and cancer tissues (IHC)	↓ FNDC5/irisin in chromophobe renal cell carcinoma	[104]
Healthy humans, thyroid cancer patients *N* = 160	irisin expression in healthy and cancer tissues (IHC)	↑ irisin in oncocytic papillary carcinoma ↑ irisin in anaplastic carcinoma	[105]
Non-small cell lung cancer patients *N* = 729	FNDC5/irisin expression in cancer tissues (IHC and RT-PCR)	↓ FNDC5/irisin levels in NSCLC tissue ↑ FNDC5/irisin levels in stromal fibroblasts	[80]

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
