# Peer review of "Current Evidence of the Role of the Myokine Irisin in Cancer"

_cancers, 2021, doi:10.3390/cancers13112628_

Round 1

Reviewer 1 Report

The manusript focuses on an interesting topic and is a solid summary of research conducted since 2012 on the involvement of irisin in various types of cancer. The title does not fully reflect the nature of the work. There is definitely a lack of emphasis on anticancer effects. Perhaps it would be worth modifying the title a bit. Work requires some corrections.

1)  Introduction – from line 35 to 80 - there is too much of information in this part of the cancer pathways  disorders. In my opinion, the information in this section should be moved to Chapter 3. Line 179-181 the functions of irisin should be moved to introduction.

2) Fig. 1 -  The figure is simplified. It is very important that irisin expression has been detected in various tissues, including healthy and neoplastic ones. It is not known whether irisin works after being released from the muscles - the data from the Elisa are not always consistent - whether it works simply released in cells such as adipocytes, nervous system or cancer cells. Fig 2 - The figure should contain the data of size, isoforms e.g. available in the Uniprot database

3) 2.1 - There is a lack of basic information about the gene encoding irisin and this information should be added before the factors influence the expression of the FNDC5 gene. What with kodon start, which is different between species. Then the resulting protein should be described and post-translational modifications, such as irisin cleavage and glycosylation.

4) This is only one part of publication ref 48 Boström et al. , which it associated with the receptor

 ‘Second, the cleaved and secreted portion of FNDC5, the hormone irisin, is highly conserved in all mammalian species sequenced. Mouse and human irisin are 100% identical, compared to 85% identity for insulin, 90% for glucagon and 83% identity for leptin. This certainly implies a highly conserved function that is likely to be mediated by a cell surface receptor. The identity of such a receptor is not yet known’

Why authors described in line 121 FNDC5 as protein which play a role as receptor based on 48 reference?

5)  Due to the large amount of information regarding e.g. pathways, factors or cell lines (which are added only in parentheses and without description) at work, it would be very advisable to add a list of abbreviations that the authors did not provide

6) line 139 - calmodulin is involved in smooth muscle contraction and how irisin expression via PGC1 is triggered in skeletal muscle fibers.It should be better described. If it depends on the ATP decrease, please describe it, of course, based on the literature.

7) 144 line – please add description that e.g C2C12 is a murine skeletal cell line and this information is from in vitro in whole manuscript

8) Line 136-138 - On the basis of recent data it appears that PGC1α is not sufficient to regulate expression. Please notice it and describe it in publications based on latest data than the original, most source work by Boström.

 e.g. Wrann et al. - ‘PGC-1α is a transcriptional co-activator, meaning it does not bind to the DNA itself but interacts with transcription factors to execute its effects on gene expression (Spiegelman,2007). The orphan nuclear receptor estrogen-related receptor alpha (ERRα; also known asNR3B1) is a central metabolic regulator (Giguere et al., 1988; Luo et al., 2003) and a very important interactor with PGC-1α (Laganiere et al., 2004; Mootha et al., 2004 al., 2004). The interaction of Errα with PGC-1α has been best studied in skeletal muscle, where it is required for mitochondrial biogenesis, induction of angiogenesis, oxidative metabolism, and oxidative muscle fibers (Arany et al., 2008; Mootha et al., 2004; Schreiberet al., 2004)’

9) line 163 – Why irisin is observed in IHC on archival samples if ‘the half-life of endogenously produced irisin in human is also less than an hour’ ?

10) The authors of ref 67 - Fan et al present the results of RT-PCR of the lung cancer lines in comparison to the normal ones but what with Nowinska et al in which authors indicted expression in lung cell A549, NCI-H1703, NCI-H522 via RT-PCR, Western-blot and Immunofluorescence in comparison to normal lung fibroblasts. The authors did not include in vitro model from this study.

11) line 282 ref 75 Please add more information about the investigation in this research on glioblastoma  what is the role of cancer-associated adipocytes ‘irisin inhibited glioblastoma cell invasion by upregulating TFPI-2 and even reversed the aggressive tumor phenotype promoted by co-cultivation with cancer-associated adipocytes ‘ (this is quite interesting in the context of the cited studies 76 where FNDC5 mRNA was higher in the adipose tissue of mice with gastric cancer) Please add infromation that this are cancer-associated adipocytes

12) The authors should add “Serum Levels of Irisin and Omentin-1 in Breast Neoplasms and Their Association with Tumor Histology”. Panagiotou G, Triantafyllidou S, Tarlatzis BC, Papakonstantinou E. – In this publication there are correlations with Elisa levels of irisin and estrogen receptors status and Ki-67 antigen in breast cancer

13) Part 3.2.2 should be more structured, because there is a lot of information about research using different techniques and different types of cancer. It can distinguish parts based on the type of analysis – e.g. ELISA, IHC  and RT-PCR

14)Table 3 - it is very important how big the group was, the results of a group of about 100 will be more reliable than, for example, 20 patients. Please add the size of the research groups in Table 3

15) animal model - humans have a different start codon than animals in the FNDC5 gene ‘The human gene however, lacks a regular start codon and is likely to use a non-AUG-initiated N-terminal extension. This feature is unique to humans. Even chimpanzee and gorilla, like any other mammal, possess a regular AUG-signal at this position indicating that the AUA-signal evolved rather recently ’ Komolka K et al PLoS One. 2014; 9(1): e88060., this may affect the research in an animal model and next one which is important for Elisa test - Jedrychowski et al. Detection and Quantitation of Circulating Human Irisin by Tandem Mass Spectrometry Cell Metab. 2015 Oct 6; 22(4): 734–740.This should be added in conclusions (like in case recombinant non-glycosylated irisin).

minor mistakes:

  • Line 303 is irisisn … should be irisin
  • line 322 – ‘ 1 unit of increase in irisin levels ..’ What unit? Please specify
  • line 345 – ‘ the mRNA levels were strongly correlated with the inflammation markers tumor necrosis factor ….’ Please specify positively or negatively correlated?
  • Table 3 irisin levels in breast [80];  measured   in liver - remove the space

Author Response

Reviewer 1

The manuscript focuses on an interesting topic and is a solid summary of research conducted since 2012 on the involvement of irisin in various types of cancer. The title does not fully reflect the nature of the work. There is definitely a lack of emphasis on anticancer effects. Perhaps it would be worth modifying the title a bit. Work requires some corrections.

We thank reviewer 1 for all the comments. We agree that the existing literature up to now does not provide very strong evidence of anticancer effects of irisin especially in human cancer patients and we have modified the title from  “Current evidence of the anticancer effects of the myokine irisin” to  “Current evidence of the role of the myokine irisin in cancer”.

1)  Introduction – from line 35 to 80 - there is too much of information in this part of the cancer pathways disorders. In my opinion, the information in this section should be moved to Chapter 3. Line 179-181 the functions of irisin should be moved to introduction.

We thank the reviewer for this comment. We followed the reviewer’s suggestions and the suggestions from the editorial office and modified the introduction.

2) Fig. 1 - The figure is simplified. It is very important that irisin expression has been detected in various tissues, including healthy and neoplastic ones. It is not known whether irisin works after being released from the muscles - the data from the Elisa are not always consistent - whether it works simply released in cells such as adipocytes, nervous system or cancer cells. Fig 2 - The figure should contain the data of size, isoforms e.g., available in the Uniprot database

We agree that the data from ELISA assays are not consistent, and it is not clear whether irisin works after being released by muscle. We have added this information in the revised manuscript.

Irisin expression is detected in various tissues, both healthy and neoplastic, and we have included this in the text.

We agree that figure 1 is simplified. It is believed that myokines after being released from exercised/contracted muscle enter the systemic circulation and delivered in different tissues exerting their effects.  We created/designed a simplified figure 1 as we aimed to depict/ convey a message regarding myokines in general (and not just irisin). For this reason, we have not modified figure 1.

The other reviewer (reviewer 2) suggested to include/reference the recent review on irisin by Maak et al.  (Endocrine Reviews 2021, PMID 33493316).  This is an excellent recent review that provides a lot of information on irisin gene, start codon in humans, irisin expression and serum detection.  We have referenced it in our manuscript. We have also cited two additional manuscripts by Albrecht et al. (Irisin: Still chasing shadows. Mol Metab 2020 Apr;34:124-135 and Irisin - a Myth Rather than an Exercise-Inducible Myokine. Sci. Rep. 2015). These 3 cited manuscripts provide a lot of information and evidence surrounding the controversies of irisin expression and detection. We have modified our manuscript to make the reader aware of these controversies.

Given the already existed/published studies by Maak’s group, that give a lot of information on irisin expression, we have only slightly modified figure 2 (included the size of irisin). 

3) 2.1 - There is a lack of basic information about the gene encoding irisin and this information should be added before the factors influence the expression of the FNDC5 gene. What with kodon start, which is different between species. Then the resulting protein should be described and post-translational modifications, such as irisin cleavage and glycosylation.

We thank the reviewer for this comment. Following the reviewer’s suggestion, we have modified the section 2.1 of our manuscript. The following information is included in section 2.1 of the revised manuscript.

“ Irisin is part of the fibronectin type III domain containing 5 (FNDC5) protein. The human FNDC5 gene has a different start codon (ATA) compared to other species such as mouse or rat (ATG) [49] and this ATA start codon is associated with low expression efficiency [50]. The reader is recommended to see recent reviews and studies by Maak et al [48] and Albrecht et al [51,52] for more information and the controversies surrounding irisin expression and detection. In a recent study Albrecht et al found  that from the three annotated human FNDC5 transcripts (T1,T2,T3), only one was expressed in human skeletal muscle [52]. Two additional transcripts (T4,T5) were found [52], indicating FNDC5 transcript diversity in human tissues.”

 “Irisin is a 112 amino acid peptide with a molecular weight of 12 KDa [53]. The amino acid sequence of irisin is highly conserved in all sequenced mammalian species and mouse and human irisin are 100% identical.”

“Post-translational modification includes N-glycosylation of FNDC5 and irisin [67–69]. Two N-glycosylation sites located at Asn-7 and Asn-52 residues of irisin have been identified and early evidence indicate an important role of glycosylation on biological activity and function [67].”

4) This is only one part of publication ref 48 Boström et al. , which it associated with the receptor

 ‘Second, the cleaved and secreted portion of FNDC5, the hormone irisin, is highly conserved in all mammalian species sequenced. Mouse and human irisin are 100% identical, compared to 85% identity for insulin, 90% for glucagon and 83% identity for leptin. This certainly implies a highly conserved function that is likely to be mediated by a cell surface receptor. The identity of such a receptor is not yet known’

Why authors described in line 121 FNDC5 as protein which play a role as receptor based on 48 reference?

Thank you very much for pointing out this mistake. Following the reviewer’s suggestion, we have made this correction in the text.

We recognize that in the paper reporting the initial discovery of FNDC5 and FNDC4, Teufel et al (Teufel A, Malik N, Mukhopadhyay M, Westphal H. Frcp1 and Frcp2, two novel fibronectin type III repeat containing genes. Gene. 2002;297(1-2):79-83.) suggested that they “are likely receptors of an as yet to be identified ligand.” We did not reference Teufel et al as we think is not required following the modification we made.

5)  Due to the large amount of information regarding e.g. pathways, factors or cell lines (which are added only in parentheses and without description) at work, it would be very advisable to add a list of abbreviations that the authors did not provide.

We followed the reviewer’s suggestion and the Author Instructions by the “cancers” editorial office and have defined the abbreviations the first time they appear.

“Acronyms/Abbreviations/Initialisms should be defined the first time they appear in each of three sections: the abstract; the main text; the first figure or table. When defined for the first time, the acronym/abbreviation/initialism should be added in parentheses after the written-out form.”

6) line 139 - calmodulin is involved in smooth muscle contraction and how irisin expression via PGC1 is triggered in skeletal muscle fibers.It should be better described. If it depends on the ATP decrease, please describe it, of course, based on the literature.

We have modified this section following the reviewer’s suggestion.

“In addition, muscle contraction reduces the cellular ATP levels resulting to increased AMP/ATP ratio and activation of the metabolic energy sensor AMP-dependent kinase (AMPK). AMPK directly phosphorylates PGC1α at threonine-177 and serine-538 [62]. PGC1α phosphorylation leads to increased expression of FNDC5 and irisin (Figure 2B) [53,63–65].”

7) 144 line – please add description that e.g C2C12 is a murine skeletal cell line and this information is from in vitro in whole manuscript

We have included that C2C12 cells are murine derived and that the information is from an in vitro study.

8) Line 136-138 - On the basis of recent data it appears that PGC1α is not sufficient to regulate expression. Please notice it and describe it in publications based on latest data than the original, most source work by Boström.

 e.g. Wrann et al. - ‘PGC-1α is a transcriptional co-activator, meaning it does not bind to the DNA itself but interacts with transcription factors to execute its effects on gene expression (Spiegelman,2007). The orphan nuclear receptor estrogen-related receptor alpha (ERRα; also known asNR3B1) is a central metabolic regulator (Giguere et al., 1988; Luo et al., 2003) and a very important interactor with PGC-1α (Laganiere et al., 2004; Mootha et al., 2004 al., 2004). The interaction of Errα with PGC-1α has been best studied in skeletal muscle, where it is required for mitochondrial biogenesis, induction of angiogenesis, oxidative metabolism, and oxidative muscle fibers (Arany et al., 2008; Mootha et al., 2004; Schreiberet al., 2004)’

We thank the reviewer for this comment. We have added the suggested information into our manuscript, as requested by the reviewer.

“The increased expression of FNDC5 and irisin in response to muscle contraction is thought to be mediated by the transcriptional coactivator peroxisome proliferator-activated receptor-γ coactivator 1α (PGC1α) [53,55]. However, evidence has shown that PGC1α is a transcriptional co-activator and does not bind to DNA itself but instead interacts with transcription factors to effect gene expression [56]. The orphan nuclear receptor estrogen-related receptor alpha (ERRα, also known as NR3B1) is a central metabolic regulator [57,58] and was found to interact with PGC1α to regulate FNDC5 gene expression [59–61]. The interaction between ERRα and PGC1α has been studied in skeletal muscle and suggests that PGC1α alone is not sufficient to regulate FNDC5 gene expression.”

9) line 163 – Why irisin is observed in IHC on archival samples if ‘the half-life of endogenously produced irisin in human is also less than an hour’ ?

Immunohistochemistry (IHC) involves the use of antibodies to detect a protein of interest (in this case irisin) in samples/tissues that have been fixed. Irisin levels detected by IHC could reflect endogenously/tissue expressed irisin or irisin that arrived to that specific tissue through the circulation. The studies that describe detection of irisin in cancer tissues were performed utilizing tumor tissue removed from the patient by surgery. The handling of the tumor tissue, time to fixation and the specificity of the different antibodies used could affect the data.

 We have added all these points in the revised manuscript.

10) The authors of ref 67 - Fan et al present the results of RT-PCR of the lung cancer lines in comparison to the normal ones but what with Nowinska et al in which authors indicted expression in lung cell A549, NCI-H1703, NCI-H522 via RT-PCR, Western-blot and Immunofluorescence in comparison to normal lung fibroblasts. The authors did not include in vitro model from this study.

We thank the reviewer for this comment. The in vitro model from the Nowinska et al. study has been added to both section 3.1 - anticancer effects of irisin: in vitro evidence and table 1.

“Additionally, RT-PCR analysis showed that NCI-1703 and NCI-H522 lung cancer cells had a higher expression of FNDC5 mRNA and higher levels of irisin protein, detected by Western-blot and immunofluorescence, compared to normal lung fibroblasts [76].”

11) line 282 ref 75 Please add more information about the investigation in this research on glioblastoma  what is the role of cancer-associated adipocytes ‘irisin inhibited glioblastoma cell invasion by upregulating TFPI-2 and even reversed the aggressive tumor phenotype promoted by co-cultivation with cancer-associated adipocytes ‘ (this is quite interesting in the context of the cited studies 76 where FNDC5 mRNA was higher in the adipose tissue of mice with gastric cancer) Please add information that this are cancer-associated adipocytes

We thank the reviewer for this comment. We have modified our manuscript accordingly. We have provided more information of both the in vitro and in vivo components of the study by Huang et al.

“Huang et al exposed U-87 MG, T98G, and LN-18 glioblastoma cells to 1 mM irisin and found a significant inhibition of cell proliferation that was associated with a G2/M cell cycle arrest and increased levels of p21 [89], (Table 1). Irisin treatment also inhibited cell invasion by upregulating the mRNA and protein expression of tissue factor pathway inhibitor-2 (TFPI-2). Co-culture of glioma (U-87 MG) cells with adipocytes (3T3L1) induced an aggressive phenotype as seen by their enhanced invasion ability. However co-culture of glioma cells with irisin-treated adipocytes reduced invasion indicating reversion of this aggressive phenotype  [89]. Examination of the culture media revealed reduced levels of several invasion-related proteases in media from glioma cells co-cultured with irisin-treated adipocytes compared to media from glioma cells co-cultured with adipocytes. These reduced levels of the invasion-related proteases, responsible for extracellular matrix (ECM) degradation, may explain the reduced invasion ability of glioma cells. These in vitro data raise the possibility that adipocytes surrounding tumors may influence cancer cell aggressiveness in vivo.”

 “Administration of irisin (20 mg/day for 2 weeks) by injection to animals xenografted with U-87 MG human glioma cells resulted in reduced tumor volume [89], (Table 2). Additionally, the use of radiolabeled irisin and micro PET/CT-fused imaging showed significant accumulation in tumors indicating a potential of irisin to target cancer cells in vivo [89]. Furthermore, slices of tumors were stained for adiponectin (used as a marker for adipocyte staining) and the data showed that the regions with strong radioactive irisin signal had positive adiponectin staining indicating that irisin and adipocytes were co-localized [89]. Interestingly, only adipocytes in tumors and areas of invasion exhibited enhanced irisin uptake [89]. These data are very interesting and suggest that irisin may target tumors in vivo by influencing/modulating cancer-associated adipocytes.”

12) The authors should add “Serum Levels of Irisin and Omentin-1 in Breast Neoplasms and Their Association with Tumor Histology”. Panagiotou G, Triantafyllidou S, Tarlatzis BC, Papakonstantinou E. – In this publication there are correlations with Elisa levels of irisin and estrogen receptors status and Ki-67 antigen in breast cancer

We thank the reviewer for this suggestion. The study by Panagiotou et al. is now include in our revised manuscript.

13) Part 3.2.2 should be more structured, because there is a lot of information about research using different techniques and different types of cancer. It can distinguish parts based on the type of analysis – e.g. ELISA, IHC and RT-PCR

 We have followed the reviewer’s suggestion and have structured the Part 3.2.2 of our manuscript.

 We have created 2 tables

Table 3.1 Role of Irisin in cancer: in vivo evidence from human studies measuring serum irisin levels

Table 3. 2 Role of Irisin in cancer: in vivo evidence from human studies measuring irisin in cancer tissues

The type of analysis e.g. ELISA, IHC and RT-PCR used to measure irisin levels is included in the revised manuscript (text and tables).

14)Table 3 - it is very important how big the group was, the results of a group of about 100 will be more reliable than, for example, 20 patients. Please add the size of the research groups in Table 3.

We have included the sample size of each study in Table 3, as the reviewer suggested.

15) animal model - humans have a different start codon than animals in the FNDC5 gene ‘The human gene however, lacks a regular start codon and is likely to use a non-AUG-initiated N-terminal extension. This feature is unique to humans. Even chimpanzee and gorilla, like any other mammal, possess a regular AUG-signal at this position indicating that the AUA-signal evolved rather recently ’ Komolka K et al PLoS One. 2014; 9(1): e88060., this may affect the research in an animal model and next one which is important for Elisa test - Jedrychowski et al. Detection and Quantitation of Circulating Human Irisin by Tandem Mass Spectrometry Cell Metab. 2015 Oct 6; 22(4): 734–740.This should be added in conclusions (like in case recombinant non-glycosylated irisin).

We thank the reviewer for this comment. We have added additional information into the conclusion of our manuscript.

“Another important issue is that the irisin concentrations used in in vitro studies are in nM range and much higher that the concentrations found in vivo.”

“However, the reliability and validity of the commercially available ELISA assays [48,51,52], utilized in most of the studies to measure irisin in human serum, is questionable and should be resolved. Hopefully more studies utilizing mass spectrometry, the gold standard, will be performed in the future to assess irisin in serum of healthy individuals and cancer patients.”

“Although FNDC5/irisin expression has been detected in various tissues, including healthy and neoplastic ones controversies exist [48,51,52] and future studies should resolve them and provide a clearer picture of irisin expression. Many studies have utilized IHC to detect irisin protein levels in healthy and tumor tissues. IHC involves the use of antibodies to detect a protein of interest (in this case irisin) in samples/tissues that have been fixed. Other studies have utilized western blotting to examine irisin levels in different tissues. Irisin levels in different tissues, detected by IHC or western blotting, could reflect endogenously/tissue expressed irisin or irisin that arrived to that specific tissue through the circulation. The studies that describe detection of irisin in cancer tissues were performed utilizing tumor tissue removed from the patient by surgery. The handling of the tumor tissue, time to fixation and the specificity of the different antibodies used could affect the data. All the current controversies surrounding the detection of irisin gene and protein expression in tissues should be resolved.”

minor mistakes:

  • Line 303 is irisisn … should be irisin
  • line 322 – ‘ 1 unit of increase in irisin levels ..’ What unit? Please specify
  • line 345 – ‘ the mRNA levels were strongly correlated with the inflammation markers tumor necrosis factor ….’ Please specify positively or negatively correlated?
  • Table 3 irisin levels in breast [80];  measured   in liver - remove the space

All minor suggestions have been addressed and corrected within the text.

Reviewer 2

The paper by Evangelia Tsiani et al. summarized studies about relationship between irisin and various kinds of cancers. The antitumor effect by irisin has been reviewed in 2019 (PMID: 31815594), but new publications are included in this paper.  The author reported that in vitro studies had shown that treatment of various cancer cells with irisin resulted in inhibition of cell proliferation, survival, migration / invasion and induced apoptosis, by affecting key proliferative and anti-apoptotic signaling pathways. However, the effects of irisin in humans remains unclear and results of serum irisin levels in cancer patients and levels of irisin in cancer tissues are controversial. The author indicated the possibility of irisin as cancer diagnosis, and/or treatment biomarkers, but concluded that more studies were required before irisin could be implemented as biomarkers. I have some comments as listed below.

Major comments:

  1. The authors described that the serum levels of irisin in human are not consistent due to the difference in the methods of irisin detection assay. This would be one of the reasons but some reports indicated the possibility of non-specific binding of the irisin antibodies to other plasma/serum proteins (Endocrine Reviews 2021, PMID 33493316). It would be better to cite this paper in this review.

We thank the reviewer for this comment.  We are grateful to this reviewer for point to us the recently published review on irisin by Maak et al ((Endocrine Reviews 2021, PMID 33493316).  Following this reviewer’s suggestion, we have cited the review manuscript by Maak et al.

We have also cited two additional manuscripts by Albrecht et al. (Irisin: Still chasing shadows. Mol Metab 2020 Apr;34:124-135 and Irisin - a Myth Rather than an Exercise-Inducible Myokine. Sci. Rep. 2015). These 3 cited manuscripts provide a lot of information and evidence surrounding the controversies of irisin expression and detection. We have modified our manuscript to make the reader aware of these controversies.

We also included the non-specific binding of irisin antibodies to other plasma proteins as another possible explanation as to why serum levels in humans are not consistent.

  1. The authors described that irisin had anticancer effects in vitro by the inhibition of the PI3K/Akt signaling cascade in some cancers. However, irisin promotes proliferation, migration, invasion via PI3K pathwayï¼»REF70ï¼½. Does the author have any idea about this contradictory evidence?

We thank the reviewer for this comment. We mention this discrepancy in section 3.1. The reason for this contradictory evidence may be cell and tissue specific effects of irisin.

“… another study utilizing hepatocellular carcinoma cells showed increased proliferation, migration, and invasion with irisin treatment [71] that was associated with activation of the PI3K/Akt signaling pathway. The existence of these contradictory evidence point to the requirement of more research and in-depth investigation of the biological effects and the role of irisin in tissue homeostasis. It may also suggest that the effects of irisin are cell and tissue specific. “

Minor comments:

  1. The following errors should be corrected.

In Table 3,ï¼»REF 92ï¼½

Serum irisin levels in “gastric “cancer patients

In Table 3,ï¼»REF81ï¼½

In the original paper, not serum but “hepatic mRNA expression” of FNDC5/Irisin is evaluated.

We thank the reviewer for these comments. All minor suggestions have been addressed and corrected within the text.

  1. In Figure 3, the author indicated that PI3K, Akt, Bcl-2, NF-kB, STAT3, mTOR were all inhibited by irisin. Whether irisin inhibited all of them directly or not? If it did not directly, please explain more detailed pathway.

We do not know if the inhibition of PI3K, Akt, BCL-2, NF-kB, STAT-3 and mTOR by irisin is direct or indirect. It is possible that irisin directly inhibits them or indirectly modulates upstream regulators. For example, the inhibition of mTOR may be due to the activation of the cellular energy sensor AMPK, an upstream regulator (inhibitor) of mTOR.  

“It is not known if the inhibition of PI3K, Akt, BCL-2, NF-kB, STAT3 and mTOR by irisin is direct or indirect. It is possible that irisin directly inhibits them or indirectly modulates upstream regulators. For example, the inhibition of mTOR may be due to the activation of the cellular energy sensor AMPK, an upstream regulator (inhibitor) of mTOR.”

Reviewer 2 Report

The paper by Evangelia Tsiani et al. summarized studies about relationship between irisin and various kinds of cancers. The antitumor effect by irisin has been reviewed in 2019 (PMID: 31815594), but new publications are included in this paper.  The author reported that in vitro studies had shown that treatment of various cancer cells with irisin resulted in inhibition of cell proliferation, survival, migration / invasion and induced apoptosis, by affecting key proliferative and anti-apoptotic signaling pathways. However, the effects of irisin in humans remains unclear and results of serum irisin levels in cancer patients and levels of irisin in cancer tissues are controversial. The author indicated the possibility of irisin as cancer diagnosis, and/or treatment biomarkers, but concluded that more studies were required before irisin could be implemented as biomarkers. I have some comments as listed below.

Major comments:

  1. The authors described that the serum levels of irisin in human are not consistent due to the difference in the methods of irisin detection assay. This would be one of the reasons but some reports indicated the possibility of non-specific binding of the irisin antibodies to other plasma/serum proteins (Endocrine Reviews 2021, PMID 33493316). It would be better to cite this paper in this review.

  1. The authors described that irisin had anticancer effects in vitro by the inhibition of the PI3K/Akt signaling cascade in some cancers. However, irisin promotes proliferation, migration, invasion via PI3K pathwayï¼»REF70ï¼½. Does the author have any idea about this contradictory evidence?

Minor comments:

  1. The following errors should be corrected.

In Table 3,ï¼»REF 92ï¼½

Serum irisin levels in “gastric “cancer patients

In Table 3,ï¼»REF81ï¼½

In the original paper, not serum but “hepatic mRNA expression” of FNDC5/Irisin is evaluated.

  1. In Figure 3, the author indicated that PI3K, Akt, Bcl-2, NF-kB, STAT3, mTOR were all inhibited by irisin. Whether irisin inhibited all of them directly or not? If it did not directly, please explain more detailed pathway.

Author Response

Reviewer 2

The paper by Evangelia Tsiani et al. summarized studies about relationship between irisin and various kinds of cancers. The antitumor effect by irisin has been reviewed in 2019 (PMID: 31815594), but new publications are included in this paper.  The author reported that in vitro studies had shown that treatment of various cancer cells with irisin resulted in inhibition of cell proliferation, survival, migration / invasion and induced apoptosis, by affecting key proliferative and anti-apoptotic signaling pathways. However, the effects of irisin in humans remains unclear and results of serum irisin levels in cancer patients and levels of irisin in cancer tissues are controversial. The author indicated the possibility of irisin as cancer diagnosis, and/or treatment biomarkers, but concluded that more studies were required before irisin could be implemented as biomarkers. I have some comments as listed below.

Major comments:

  1. The authors described that the serum levels of irisin in human are not consistent due to the difference in the methods of irisin detection assay. This would be one of the reasons but some reports indicated the possibility of non-specific binding of the irisin antibodies to other plasma/serum proteins (Endocrine Reviews 2021, PMID 33493316). It would be better to cite this paper in this review.

We thank the reviewer for this comment.  We are grateful to this reviewer for point to us the recently published review on irisin by Maak et al ((Endocrine Reviews 2021, PMID 33493316).  Following this reviewer’s suggestion, we have cited the review manuscript by Maak et al.

We have also cited two additional manuscripts by Albrecht et al. (Irisin: Still chasing shadows. Mol Metab 2020 Apr;34:124-135 and Irisin - a Myth Rather than an Exercise-Inducible Myokine. Sci. Rep. 2015). These 3 cited manuscripts provide a lot of information and evidence surrounding the controversies of irisin expression and detection. We have modified our manuscript to make the reader aware of these controversies.

We also included the non-specific binding of irisin antibodies to other plasma proteins as another possible explanation as to why serum levels in humans are not consistent.

  1. The authors described that irisin had anticancer effects in vitro by the inhibition of the PI3K/Akt signaling cascade in some cancers. However, irisin promotes proliferation, migration, invasion via PI3K pathwayï¼»REF70ï¼½. Does the author have any idea about this contradictory evidence?

We thank the reviewer for this comment. We mention this discrepancy in section 3.1. The reason for this contradictory evidence may be cell and tissue specific effects of irisin.

“… another study utilizing hepatocellular carcinoma cells showed increased proliferation, migration, and invasion with irisin treatment [71] that was associated with activation of the PI3K/Akt signaling pathway. The existence of these contradictory evidence point to the requirement of more research and in-depth investigation of the biological effects and the role of irisin in tissue homeostasis. It may also suggest that the effects of irisin are cell and tissue specific. “

Minor comments:

  1. The following errors should be corrected.

In Table 3,ï¼»REF 92ï¼½

Serum irisin levels in “gastric “cancer patients

In Table 3,ï¼»REF81ï¼½

In the original paper, not serum but “hepatic mRNA expression” of FNDC5/Irisin is evaluated.

We thank the reviewer for these comments. All minor suggestions have been addressed and corrected within the text.

  1. In Figure 3, the author indicated that PI3K, Akt, Bcl-2, NF-kB, STAT3, mTOR were all inhibited by irisin. Whether irisin inhibited all of them directly or not? If it did not directly, please explain more detailed pathway.

We do not know if the inhibition of PI3K, Akt, BCL-2, NF-kB, STAT-3 and mTOR by irisin is direct or indirect. It is possible that irisin directly inhibits them or indirectly modulates upstream regulators. For example, the inhibition of mTOR may be due to the activation of the cellular energy sensor AMPK, an upstream regulator (inhibitor) of mTOR.  

“It is not known if the inhibition of PI3K, Akt, BCL-2, NF-kB, STAT3 and mTOR by irisin is direct or indirect. It is possible that irisin directly inhibits them or indirectly modulates upstream regulators. For example, the inhibition of mTOR may be due to the activation of the cellular energy sensor AMPK, an upstream regulator (inhibitor) of mTOR.”

Reviewer 1

The manuscript focuses on an interesting topic and is a solid summary of research conducted since 2012 on the involvement of irisin in various types of cancer. The title does not fully reflect the nature of the work. There is definitely a lack of emphasis on anticancer effects. Perhaps it would be worth modifying the title a bit. Work requires some corrections.

We thank reviewer 1 for all the comments. We agree that the existing literature up to now does not provide very strong evidence of anticancer effects of irisin especially in human cancer patients and we have modified the title from  “Current evidence of the anticancer effects of the myokine irisin” to  “Current evidence of the role of the myokine irisin in cancer”.

1)  Introduction – from line 35 to 80 - there is too much of information in this part of the cancer pathways disorders. In my opinion, the information in this section should be moved to Chapter 3. Line 179-181 the functions of irisin should be moved to introduction.

We thank the reviewer for this comment. We followed the reviewer’s suggestions and the suggestions from the editorial office and modified the introduction.

2) Fig. 1 - The figure is simplified. It is very important that irisin expression has been detected in various tissues, including healthy and neoplastic ones. It is not known whether irisin works after being released from the muscles - the data from the Elisa are not always consistent - whether it works simply released in cells such as adipocytes, nervous system or cancer cells. Fig 2 - The figure should contain the data of size, isoforms e.g., available in the Uniprot database

We agree that the data from ELISA assays are not consistent, and it is not clear whether irisin works after being released by muscle. We have added this information in the revised manuscript.

Irisin expression is detected in various tissues, both healthy and neoplastic, and we have included this in the text.

We agree that figure 1 is simplified. It is believed that myokines after being released from exercised/contracted muscle enter the systemic circulation and delivered in different tissues exerting their effects.  We created/designed a simplified figure 1 as we aimed to depict/ convey a message regarding myokines in general (and not just irisin). For this reason, we have not modified figure 1.

The other reviewer (reviewer 2) suggested to include/reference the recent review on irisin by Maak et al.  (Endocrine Reviews 2021, PMID 33493316).  This is an excellent recent review that provides a lot of information on irisin gene, start codon in humans, irisin expression and serum detection.  We have referenced it in our manuscript. We have also cited two additional manuscripts by Albrecht et al. (Irisin: Still chasing shadows. Mol Metab 2020 Apr;34:124-135 and Irisin - a Myth Rather than an Exercise-Inducible Myokine. Sci. Rep. 2015). These 3 cited manuscripts provide a lot of information and evidence surrounding the controversies of irisin expression and detection. We have modified our manuscript to make the reader aware of these controversies.

Given the already existed/published studies by Maak’s group, that give a lot of information on irisin expression, we have only slightly modified figure 2 (included the size of irisin). 

3) 2.1 - There is a lack of basic information about the gene encoding irisin and this information should be added before the factors influence the expression of the FNDC5 gene. What with kodon start, which is different between species. Then the resulting protein should be described and post-translational modifications, such as irisin cleavage and glycosylation.

We thank the reviewer for this comment. Following the reviewer’s suggestion, we have modified the section 2.1 of our manuscript. The following information is included in section 2.1 of the revised manuscript.

“ Irisin is part of the fibronectin type III domain containing 5 (FNDC5) protein. The human FNDC5 gene has a different start codon (ATA) compared to other species such as mouse or rat (ATG) [49] and this ATA start codon is associated with low expression efficiency [50]. The reader is recommended to see recent reviews and studies by Maak et al [48] and Albrecht et al [51,52] for more information and the controversies surrounding irisin expression and detection. In a recent study Albrecht et al found  that from the three annotated human FNDC5 transcripts (T1,T2,T3), only one was expressed in human skeletal muscle [52]. Two additional transcripts (T4,T5) were found [52], indicating FNDC5 transcript diversity in human tissues.”

 “Irisin is a 112 amino acid peptide with a molecular weight of 12 KDa [53]. The amino acid sequence of irisin is highly conserved in all sequenced mammalian species and mouse and human irisin are 100% identical.”

“Post-translational modification includes N-glycosylation of FNDC5 and irisin [67–69]. Two N-glycosylation sites located at Asn-7 and Asn-52 residues of irisin have been identified and early evidence indicate an important role of glycosylation on biological activity and function [67].”

4) This is only one part of publication ref 48 Boström et al. , which it associated with the receptor

 ‘Second, the cleaved and secreted portion of FNDC5, the hormone irisin, is highly conserved in all mammalian species sequenced. Mouse and human irisin are 100% identical, compared to 85% identity for insulin, 90% for glucagon and 83% identity for leptin. This certainly implies a highly conserved function that is likely to be mediated by a cell surface receptor. The identity of such a receptor is not yet known’

Why authors described in line 121 FNDC5 as protein which play a role as receptor based on 48 reference?

Thank you very much for pointing out this mistake. Following the reviewer’s suggestion, we have made this correction in the text.

We recognize that in the paper reporting the initial discovery of FNDC5 and FNDC4, Teufel et al (Teufel A, Malik N, Mukhopadhyay M, Westphal H. Frcp1 and Frcp2, two novel fibronectin type III repeat containing genes. Gene. 2002;297(1-2):79-83.) suggested that they “are likely receptors of an as yet to be identified ligand.” We did not reference Teufel et al as we think is not required following the modification we made.

5)  Due to the large amount of information regarding e.g. pathways, factors or cell lines (which are added only in parentheses and without description) at work, it would be very advisable to add a list of abbreviations that the authors did not provide.

We followed the reviewer’s suggestion and the Author Instructions by the “cancers” editorial office and have defined the abbreviations the first time they appear.

“Acronyms/Abbreviations/Initialisms should be defined the first time they appear in each of three sections: the abstract; the main text; the first figure or table. When defined for the first time, the acronym/abbreviation/initialism should be added in parentheses after the written-out form.”

6) line 139 - calmodulin is involved in smooth muscle contraction and how irisin expression via PGC1 is triggered in skeletal muscle fibers.It should be better described. If it depends on the ATP decrease, please describe it, of course, based on the literature.

We have modified this section following the reviewer’s suggestion.

“In addition, muscle contraction reduces the cellular ATP levels resulting to increased AMP/ATP ratio and activation of the metabolic energy sensor AMP-dependent kinase (AMPK). AMPK directly phosphorylates PGC1α at threonine-177 and serine-538 [62]. PGC1α phosphorylation leads to increased expression of FNDC5 and irisin (Figure 2B) [53,63–65].”

7) 144 line – please add description that e.g C2C12 is a murine skeletal cell line and this information is from in vitro in whole manuscript

We have included that C2C12 cells are murine derived and that the information is from an in vitro study.

8) Line 136-138 - On the basis of recent data it appears that PGC1α is not sufficient to regulate expression. Please notice it and describe it in publications based on latest data than the original, most source work by Boström.

 e.g. Wrann et al. - ‘PGC-1α is a transcriptional co-activator, meaning it does not bind to the DNA itself but interacts with transcription factors to execute its effects on gene expression (Spiegelman,2007). The orphan nuclear receptor estrogen-related receptor alpha (ERRα; also known asNR3B1) is a central metabolic regulator (Giguere et al., 1988; Luo et al., 2003) and a very important interactor with PGC-1α (Laganiere et al., 2004; Mootha et al., 2004 al., 2004). The interaction of Errα with PGC-1α has been best studied in skeletal muscle, where it is required for mitochondrial biogenesis, induction of angiogenesis, oxidative metabolism, and oxidative muscle fibers (Arany et al., 2008; Mootha et al., 2004; Schreiberet al., 2004)’

We thank the reviewer for this comment. We have added the suggested information into our manuscript, as requested by the reviewer.

“The increased expression of FNDC5 and irisin in response to muscle contraction is thought to be mediated by the transcriptional coactivator peroxisome proliferator-activated receptor-γ coactivator 1α (PGC1α) [53,55]. However, evidence has shown that PGC1α is a transcriptional co-activator and does not bind to DNA itself but instead interacts with transcription factors to effect gene expression [56]. The orphan nuclear receptor estrogen-related receptor alpha (ERRα, also known as NR3B1) is a central metabolic regulator [57,58] and was found to interact with PGC1α to regulate FNDC5 gene expression [59–61]. The interaction between ERRα and PGC1α has been studied in skeletal muscle and suggests that PGC1α alone is not sufficient to regulate FNDC5 gene expression.”

9) line 163 – Why irisin is observed in IHC on archival samples if ‘the half-life of endogenously produced irisin in human is also less than an hour’ ?

Immunohistochemistry (IHC) involves the use of antibodies to detect a protein of interest (in this case irisin) in samples/tissues that have been fixed. Irisin levels detected by IHC could reflect endogenously/tissue expressed irisin or irisin that arrived to that specific tissue through the circulation. The studies that describe detection of irisin in cancer tissues were performed utilizing tumor tissue removed from the patient by surgery. The handling of the tumor tissue, time to fixation and the specificity of the different antibodies used could affect the data.

 We have added all these points in the revised manuscript.

10) The authors of ref 67 - Fan et al present the results of RT-PCR of the lung cancer lines in comparison to the normal ones but what with Nowinska et al in which authors indicted expression in lung cell A549, NCI-H1703, NCI-H522 via RT-PCR, Western-blot and Immunofluorescence in comparison to normal lung fibroblasts. The authors did not include in vitro model from this study.

We thank the reviewer for this comment. The in vitro model from the Nowinska et al. study has been added to both section 3.1 - anticancer effects of irisin: in vitro evidence and table 1.

“Additionally, RT-PCR analysis showed that NCI-1703 and NCI-H522 lung cancer cells had a higher expression of FNDC5 mRNA and higher levels of irisin protein, detected by Western-blot and immunofluorescence, compared to normal lung fibroblasts [76].”

11) line 282 ref 75 Please add more information about the investigation in this research on glioblastoma  what is the role of cancer-associated adipocytes ‘irisin inhibited glioblastoma cell invasion by upregulating TFPI-2 and even reversed the aggressive tumor phenotype promoted by co-cultivation with cancer-associated adipocytes ‘ (this is quite interesting in the context of the cited studies 76 where FNDC5 mRNA was higher in the adipose tissue of mice with gastric cancer) Please add information that this are cancer-associated adipocytes

We thank the reviewer for this comment. We have modified our manuscript accordingly. We have provided more information of both the in vitro and in vivo components of the study by Huang et al.

“Huang et al exposed U-87 MG, T98G, and LN-18 glioblastoma cells to 1 mM irisin and found a significant inhibition of cell proliferation that was associated with a G2/M cell cycle arrest and increased levels of p21 [89], (Table 1). Irisin treatment also inhibited cell invasion by upregulating the mRNA and protein expression of tissue factor pathway inhibitor-2 (TFPI-2). Co-culture of glioma (U-87 MG) cells with adipocytes (3T3L1) induced an aggressive phenotype as seen by their enhanced invasion ability. However co-culture of glioma cells with irisin-treated adipocytes reduced invasion indicating reversion of this aggressive phenotype  [89]. Examination of the culture media revealed reduced levels of several invasion-related proteases in media from glioma cells co-cultured with irisin-treated adipocytes compared to media from glioma cells co-cultured with adipocytes. These reduced levels of the invasion-related proteases, responsible for extracellular matrix (ECM) degradation, may explain the reduced invasion ability of glioma cells. These in vitro data raise the possibility that adipocytes surrounding tumors may influence cancer cell aggressiveness in vivo.”

 “Administration of irisin (20 mg/day for 2 weeks) by injection to animals xenografted with U-87 MG human glioma cells resulted in reduced tumor volume [89], (Table 2). Additionally, the use of radiolabeled irisin and micro PET/CT-fused imaging showed significant accumulation in tumors indicating a potential of irisin to target cancer cells in vivo [89]. Furthermore, slices of tumors were stained for adiponectin (used as a marker for adipocyte staining) and the data showed that the regions with strong radioactive irisin signal had positive adiponectin staining indicating that irisin and adipocytes were co-localized [89]. Interestingly, only adipocytes in tumors and areas of invasion exhibited enhanced irisin uptake [89]. These data are very interesting and suggest that irisin may target tumors in vivo by influencing/modulating cancer-associated adipocytes.”

12) The authors should add “Serum Levels of Irisin and Omentin-1 in Breast Neoplasms and Their Association with Tumor Histology”. Panagiotou G, Triantafyllidou S, Tarlatzis BC, Papakonstantinou E. – In this publication there are correlations with Elisa levels of irisin and estrogen receptors status and Ki-67 antigen in breast cancer

We thank the reviewer for this suggestion. The study by Panagiotou et al. is now include in our revised manuscript.

13) Part 3.2.2 should be more structured, because there is a lot of information about research using different techniques and different types of cancer. It can distinguish parts based on the type of analysis – e.g. ELISA, IHC and RT-PCR

 We have followed the reviewer’s suggestion and have structured the Part 3.2.2 of our manuscript.

 We have created 2 tables

Table 3.1 Role of Irisin in cancer: in vivo evidence from human studies measuring serum irisin levels

Table 3. 2 Role of Irisin in cancer: in vivo evidence from human studies measuring irisin in cancer tissues

The type of analysis e.g. ELISA, IHC and RT-PCR used to measure irisin levels is included in the revised manuscript (text and tables).

14)Table 3 - it is very important how big the group was, the results of a group of about 100 will be more reliable than, for example, 20 patients. Please add the size of the research groups in Table 3.

We have included the sample size of each study in Table 3, as the reviewer suggested.

15) animal model - humans have a different start codon than animals in the FNDC5 gene ‘The human gene however, lacks a regular start codon and is likely to use a non-AUG-initiated N-terminal extension. This feature is unique to humans. Even chimpanzee and gorilla, like any other mammal, possess a regular AUG-signal at this position indicating that the AUA-signal evolved rather recently ’ Komolka K et al PLoS One. 2014; 9(1): e88060., this may affect the research in an animal model and next one which is important for Elisa test - Jedrychowski et al. Detection and Quantitation of Circulating Human Irisin by Tandem Mass Spectrometry Cell Metab. 2015 Oct 6; 22(4): 734–740.This should be added in conclusions (like in case recombinant non-glycosylated irisin).

We thank the reviewer for this comment. We have added additional information into the conclusion of our manuscript.

“Another important issue is that the irisin concentrations used in in vitro studies are in nM range and much higher that the concentrations found in vivo.”

“However, the reliability and validity of the commercially available ELISA assays [48,51,52], utilized in most of the studies to measure irisin in human serum, is questionable and should be resolved. Hopefully more studies utilizing mass spectrometry, the gold standard, will be performed in the future to assess irisin in serum of healthy individuals and cancer patients.”

“Although FNDC5/irisin expression has been detected in various tissues, including healthy and neoplastic ones controversies exist [48,51,52] and future studies should resolve them and provide a clearer picture of irisin expression. Many studies have utilized IHC to detect irisin protein levels in healthy and tumor tissues. IHC involves the use of antibodies to detect a protein of interest (in this case irisin) in samples/tissues that have been fixed. Other studies have utilized western blotting to examine irisin levels in different tissues. Irisin levels in different tissues, detected by IHC or western blotting, could reflect endogenously/tissue expressed irisin or irisin that arrived to that specific tissue through the circulation. The studies that describe detection of irisin in cancer tissues were performed utilizing tumor tissue removed from the patient by surgery. The handling of the tumor tissue, time to fixation and the specificity of the different antibodies used could affect the data. All the current controversies surrounding the detection of irisin gene and protein expression in tissues should be resolved.”

minor mistakes:

  • Line 303 is irisisn … should be irisin
  • line 322 – ‘ 1 unit of increase in irisin levels ..’ What unit? Please specify
  • line 345 – ‘ the mRNA levels were strongly correlated with the inflammation markers tumor necrosis factor ….’ Please specify positively or negatively correlated?
  • Table 3 irisin levels in breast [80];  measured   in liver - remove the space

All minor suggestions have been addressed and corrected within the text.

Round 2

Reviewer 1 Report

accept in current form